# Mechanism of barotaxis in marine zooplankton

**Luis Alberto Bezares Calderón[1]\*, Réza Shahidi[1,2], Gáspár Jékely[1,3]\***

[1]Living Systems Institute, University of Exeter, Exeter, United Kingdom; [2]Electron Microscopy Core Facility (EMCF), Heidelberg University, Heidelberg, Germany; [3]Centre for Organismal Studies (COS), Heidelberg University, Heidelberg, Germany

**Abstract** Hydrostatic pressure is a dominant environmental cue for vertically migrating marine organisms but the physiological mechanisms of responding to pressure changes remain unclear. Here, we uncovered the cellular and circuit bases of a barokinetic response in the planktonic larva of the marine annelid *Platynereis dumerilii*. Increased pressure induced a rapid, graded, and adapting upward swimming response due to the faster beating of cilia in the head multiciliary band. By calcium imaging, we found that brain ciliary photoreceptors showed a graded response to pressure changes. The photoreceptors in animals mutant for *ciliary opsin-1* had a smaller sensory compartment and mutant larvae showed diminished pressure responses. The ciliary photoreceptors synaptically connect to the head multiciliary band via serotonergic motoneurons. Genetic inhibition of the serotonergic cells blocked pressure-dependent increases in ciliary beating. We conclude that ciliary photoreceptors function as pressure sensors and activate ciliary beating through serotonergic signalling during barokinesis.

**\*For correspondence:**
L.A.Bezares-Calderon@exeter.ac.uk (LAlbertoBC);
gaspar.jekely@cos.uni-heidelberg.de (GJ)

## eLife assessment

This **fundamental** study addresses the question of how certain zooplankton achieve barotaxis, directed locomotion in response to changes in hydraulic pressure. The authors provide **compelling** evidence that the response involves ciliary photoreceptors interacting with motoneurons. This work should be of broad interest to scientists working on mechanosensation, cilia, locomotion, and photoreceptors.

## Introduction

Hydrostatic pressure increases linearly with depth in the ocean and planktonic organisms can use it as a depth cue, which is independent of light or the time of the day (*Blaxter, 1978*). Many marine invertebrate animals have long been known to sense and respond to changes in pressure (*Knight-Jones and Qasim, 1955*; *Rice, 1964*). The response generally consists of an increase in locomotion (barokinesis) upon an increase in pressure. Such responses could help planktonic animals retain their depth either in combination with, or independent of light cues (*Forward et al., 1989*). Changes in hydrostatic pressure may additionally entrain tidal rhythms in marine animals (*Akiyama, 2004*; *Morgan, 1965*; *Naylor and Williams, 1984*). Early studies on the barokinetic response in zooplankton have not revealed if the animals respond to relative or absolute changes in pressure. The kinematics and neuronal mechanisms of pressure responses have also not been characterized in detail for any planktonic animal. The most familiar structures for sensing changes in hydrostatic pressure are gas-filled compressible vesicles such as the swim bladder in fish (*Qutob, 1963*). However, barokinetic responses are seen across many animals without any identifiable gas-filled vesicles. What structures could mediate pressure sensing in these organisms? Thus far, only a few alternative structures have been proposed for

pressure sensation. In the statocyst of the adult crab *Carcinus maenas*, millimeter-sized thread-hairs may act as a syringe plunger to sense pressure (*Fraser and Macdonald, 1994*). In dogfish, which lack a swim bladder, hair cells in the vestibular organ have been proposed to act as pressure detectors (*Fraser and Shelmerdine, 2002*). It is unknown which, if any, of these two vastly different pressure sensing mechanisms—one based on volume changes in a gas-filled vesicle and the other on deformation of sensory cilia—is used by the much smaller planktonic animals.

To understand the behavioural and neuronal mechanisms of pressure responses in marine zooplankton, we studied the planktonic ciliated larvae of the marine annelid *Platynereis dumerilii* (*Özpolat et al., 2021*). This larva uses ciliary beating to swim up and down in the water column to eventually settle on sea grass beds near coastal regions (*Gambi et al., 1992*). The sensory and neuronal bases of light-guided (*Gühmann et al., 2015*; *Randel et al., 2014*; *Verasztó et al., 2018*) and mechanically driven behaviours (*Bezares-Calderón et al., 2018*) in *Platynereis* larvae have been dissected due to the experimental tractability of this system. Its small size has allowed the entire reconstruction of the cellular and synaptic wiring map of the 3-day-old larva (*Jasek et al., 2022*; *Verasztó et al., 2024*). Here, we study *Platynereis* larvae to understand the cellular and neuronal bases of pressure sensation in zooplankton.

## Results

### *Platynereis* larvae respond to changes in hydrostatic pressure

To determine whether *Platynereis* larvae respond to changes in hydrostatic pressure, we developed a custom behavioural chamber with precise pressure control. We subjected larvae to step changes in pressure and recorded their behaviour under near-infrared illumination (*Figure 1A*; see Materials and methods). We used hydrocarbon-free compressed air to increase pressure in the chamber. We tested a range of pressure levels in randomized order from 3 to 1000 mbar (1 mbar equals to 1 cm water depth) (*Figure 1—figure supplement 1A*). We focused on 1- to 3-day-old larvae corresponding to the early and late trochophore and nectochaete stages. We used batches of >100 larvae for each experiment. Both 2- and 3-day-old larvae respond to pressure increase by swimming upwards faster and in straighter trajectories, as quantified by changes in average vertical displacement, swimming speed, the ratio of upward to downward trajectories (*Figure 1B, C*, *Figure 1—figure supplement 1B–D*; *Video 1*), and a straightness index (the net over total distance) (*Figure 1—figure supplement 1E*).

We used the maximal vertical displacement value (normalized to the mean displacement per trial prior to stimulus presentation) to compare the magnitude of responses as a function of pressure change. In both 2- and 3-day-old larvae, the responses were graded: higher pressure levels led to higher maximal vertical displacement (*Figure 1D, E*). Three-day-old larvae had a slightly lower sensitivity threshold (10–20 mbar) than 2-day-old larvae (20–30 mbar) and their response plateaued at lower pressure levels than that of 2-day-old larvae (*Figure 1B–E*). One-day-old larvae did not show a detectable response to even the largest pressure levels tested (*Figure 1—figure supplement 2A, B*). The straightness of trajectories also increased with increasing pressure changes. This was due to narrower helical swimming paths under pressure, visible in close-up videos (*Video 2* and *Video 3*). Three-day-old larvae also showed a diving response upon the release of pressure (*Figure 1B, C, F*; *Figure 1—figure supplement 1C, D*). Two-day-old larvae stopped moving upwards when the stimulus ended, but did not show an active diving response to pressure OFF.

To exclude the possibility that either the changes in the partial pressure of gases due to the use of compressed air, or the mechanical wave associated to the inflow of air caused the upward swimming behaviour, we also used a static column of water of different heights to change pressure levels (*Figure 1—figure supplement 2C*). We observed the same dependence of vertical displacement on the magnitude of pressure change in this setup (*Figure 1—figure supplement 2D, E*). Overall, our experiments uncovered a graded, saturable, and highly sensitive upward swimming ON response to increased pressure in *Platynereis* larvae and a diving OFF response (in 3-day-old larvae only).

### *Platynereis* larvae respond to relative changes of pressure

Larvae may either detect absolute pressure levels, relative changes, or the rate at which pressure changes (*Morgan, 1984*). To differentiate between these possibilities, we first exposed larvae to linear increases of pressure with rates between 0.05 and 2.6 mbar s$^{-1}$ (*Figure 1—figure supplement 3A*).

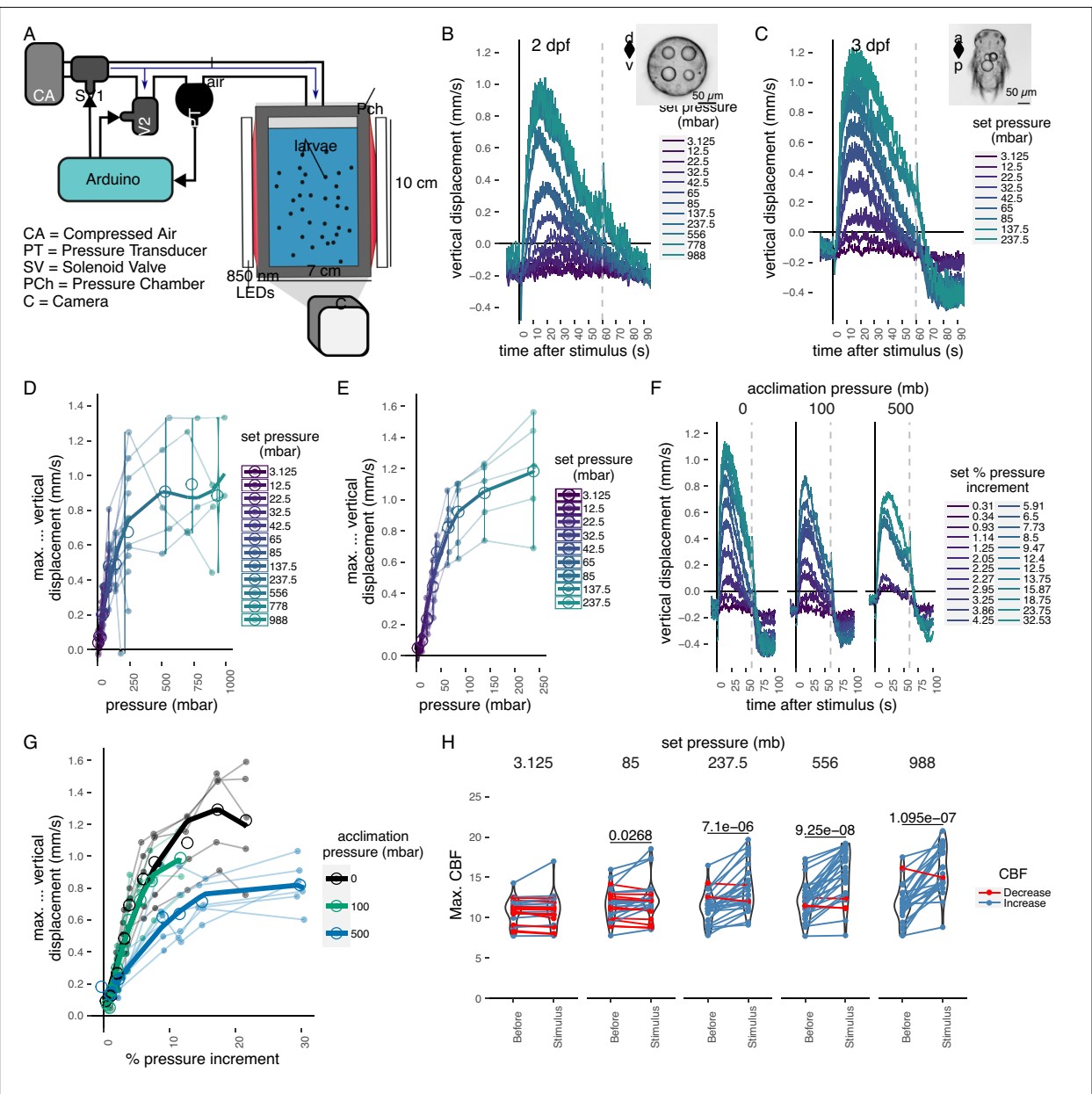

**Figure 1.** Pressure response in *Platynereis* larvae. (**A**) Schematic of the behavioural setup used to stimulate larvae with controlled pressure levels. Vertical displacement of (**B**) 2-day-old and (**C**) 3-day-old larvae (insets) as a function of time relative to different step increases in pressure. Dashed line at 60 s indicates the end of stimulation. Each data point is the average of 2–12 (**B**) or 4–5 (**C**) batches of larvae. Maximum increase in relative vertical displacement of (**D**) 2-day-old and (**E**) 3-day-old larvae for each pressure level tested. Small filled circles represent individual data points, data points from the same batch are joined by lines. Larger open circles indicate the mean across all batches. (**F**) Vertical displacement of 3-day-old larvae acclimated for ca. 10 min to either ambient pressure (0 mb, left), 100 mb (centre), or 500 mb (right) prior to the experiment. Lines are coloured by set fractional increments in pressure. (**G**) Maximum increase in relative vertical displacement of 3-day-old larvae acclimated to 0, 100, or 500 mb. The data are fitted with a saturation curve. 5 (0, 100 mb) or 6 (500 mb) batches tested in **F**–**G**. (**H**) Maximum ciliary beat frequency (CBF) that single larvae reached in the 30 s prior (category Before), or during the first 30 s (Stimulus) of the indicated increase in pressure. $N = 18$–22 larvae. Data points for the same larva are joined by lines. One-tailed paired *t*-test with Bonferroni correction testing for an increase in CBF; p-values <0.05 are shown. Error bars in **B**, **C**, and **F** show the standard error of the mean. Data in **D**, **E**, and **G** were fitted with a third (**D**, **E**) or a second (**G**) order polynomial function. *Figure 1—source data 1* (**A, D**), *Figure 1—source data 2* (**C, E**), *Figure 1—source data 3* (**F, G**), *Figure 1—source data 4* (**H**).

The online version of this article includes the following source data and figure supplement(s) for figure 1:

**Source data 1.** Swimming metrics of mixed batches of 2-day-old wildtype larvae subjected to randomized step increases in pressure.

**Source data 2.** Swimming metrics of mixed batches of 3-day-old wildtype larvae subjected to randomized step increases in pressure.

*Figure 1 continued on next page*

*Figure 1 continued*

**Source data 3.** Swimming metrics of mixed batches of 3-day-old wildtype larvae subjected to randomized step increases in pressure after acclimatization to higher than ambient pressures for 10 min.

**Source data 4.** Ciliary-band dynamics of individual 2-day-old wildtype or c-opsin-1 mutant larvae subjected to randomized step increases in pressure.

**Source data 5.** Swimming metrics of 1-day-old-larvae subjected to randomized step increases in pressure.

**Source data 6.** Swimming metrics of 3-day-old-larvae subjected to randomized step increases in pressure using a water column of different heights.

**Source data 7.** Swimming metrics of 3-day-old-larvae subjected to randomized linear increases in pressure.

**Figure supplement 1.** Quantification of swimming behaviour in *Platynereis* larvae in response to pressure.

**Figure supplement 2.** Swimming behaviour in 1-day-old-larvae to changes in pressure and in 3-day-old-larvae to the application of a hydrostatic pressure stimulus by a column of water.

**Figure supplement 3.** Quantification of swimming behaviour in 3-day-old larvae subjected to linear increases in pressure and to long-lasting periods of increased pressure.

**Figure supplement 4.** Ciliary dynamics assay.

---

We used a second-degree polynomial function for rate categories 0.3–0.7 mbar s$^{-1}$ (analysis of variance [ANOVA], p = 6.5e−3) and 0.9–1.3 mbar s$^{-1}$ (ANOVA, p = 5.8e−8), as it described the relationship between vertical displacement and pressure more accurately than a simple linear model. The difference between a linear and a second-degree polynomial fit was not significant for rates >1.3 mbar s$^{-1}$ (ANOVA, p ~ 0.1). These results suggest that larvae compensate for the increase in pressure by a corresponding increase in upward swimming when rates of pressure increase are sufficiently high.

The linear response to a gradual increase in pressure suggests that larvae detect changes in pressure, rather than absolute pressure levels. To directly address this, we acclimated 3-day-old larvae for ca. 10 min to either 100 or 500 mbar pressure above the atmospheric level. We then tested a range of randomized step increases in pressure levels (*Figure 1—figure supplement 3D*). After the acclimation period, the distribution of larvae exposed to 100 or 500 mbar was not different from the larvae kept at ambient levels (two-sided Kolmogorov–Smirnoff test, 0–100 mbar: p = 0.915, 0–500 mbar: p = 0.0863) (*Figure 1—figure supplement 3E, F*). Upon step increase, larvae reacted with graded upward swimming even if they were pre-exposed to 100 or 500 mbar (*Figure 1F*). The sensitivity decreased when larvae were acclimated to 500 mbar (*Figure 1G*). When pressure was released at the end of the increase trials, larvae pre-exposed to 500 mbar showed a downward displacement followed by an upward displacement as soon as pressure was increased back to the corresponding basal level (*Figure 1—figure supplement 3G*). The downward displacement resembled the magnitude of the diving response we observed for 3-day-old larvae (*Figure 1—figure supplement 3H*).

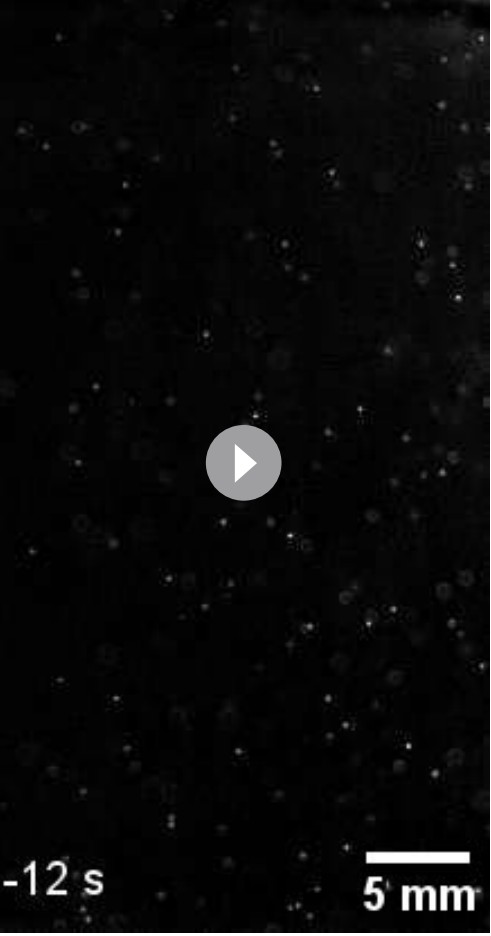

−12 s 5 mm

**Video 1.** Response of 2-day-old *Platynereis* larvae to a pressure increase of 1 bar. Larvae were placed in a pressure vessel (100 mm height) and recorded in effective darkness (with 850 nm light) with a camera placed in front of the vessel (see *Figure 1A*). Larvae swim towards the top of the chamber as soon as pressure is increased. Blue traces (10 s long) delineate swimming trajectories.

https://elifesciences.org/articles/94306/figures#video1

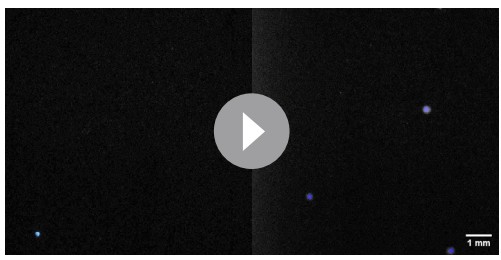

**Video 2.** Zoomed-in view of 2-day-old larvae as they respond to an increase in pressure of 85 mb (left panel) or 500 mb. Blue traces (each 4 s long) delineate swimming trajectories. The two videos are played at the same speed.

https://elifesciences.org/articles/94306/figures#video2

Our experiments suggest that *Platynereis* larvae react to relative increases in pressure in a graded manner proportional to the magnitude of the increase. The response is adaptable and occurs at very different basal pressures (0 or 500 mbar—corresponding to surface or 5 m of water depth). This hints at a pressure-gauge mechanism to regulate swimming depth by compensating for vertical movements due to down-welling currents (*Genin et al., 2005*), sinking when cilia are arrested (*Verasztó et al., 2017*) or downward swimming (e.g. during UV avoidance; *Verasztó et al., 2018*).

## Ciliary beat frequency increases with pressure

To understand the mechanism by which larvae regulate swimming in response to an increase in pressure, we analysed the effect of pressure on ciliary beating in the prototroch—the main ciliary band that propels swimming in 2- and 3-day-old *Platynereis* larvae. Individual 2-day-old larvae were tethered to a glass cuvette from the posterior end with a non-toxic glue previously used in *Platynereis* larvae (*Bezares-Calderón et al., 2018*). The cuvette was inserted into a custom-made pressure vessel placed under a microscope. We applied 60 s step increases in pressure in a randomized order and recorded ciliary beating in effective darkness (*Figure 1—figure supplement 4A, B*).

The mean ciliary beat frequency (CBF) increased as soon as a step change in pressure was applied, with larger step changes showing more noticeable increases in beat frequency (*Figure 1—figure supplement 4C*, *Video 4*). The maximum ciliary beat frequency (max. CBF) during the stimulus period showed a statistically significant increase for all but the lowest pressure steps tested relative to the period before the onset of the stimulus (85 mb p = 0.046, 237.5 mb p = 2.08E−05, 556 mb p = 8.35E−07, 988 mb p = 7.8E−07; one-tailed paired *t*-test with Bonferroni correction testing for an increase in CBF; *Figure 1H*). The related relative measure of maximum percent change (max. Δ%CBF) also showed an increase under pressure (*Figure 1—figure supplement 4D*). Overall, these data suggest that rapid upward swimming under pressure is due to an increase in the beating frequency of prototroch cilia that is proportional to the change in pressure.

## Brain ciliary photoreceptor cells show graded activation under increased pressure

To identify the pressure-sensitive cells in *Platynereis* larvae, we developed an approach to couple

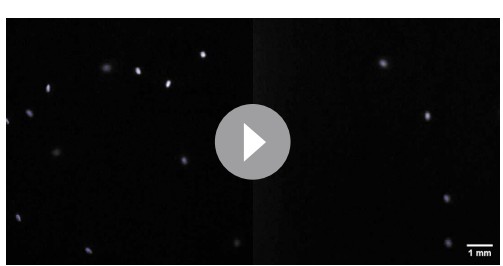

**Video 3.** Zoomed-in view of 3-day-old larvae as they respond to an increase in pressure of 32 mb (left panel) or 778 mb. Blue traces (each 4 s long) delineate swimming trajectories. The two videos are played at the same speed.

https://elifesciences.org/articles/94306/figures#video3

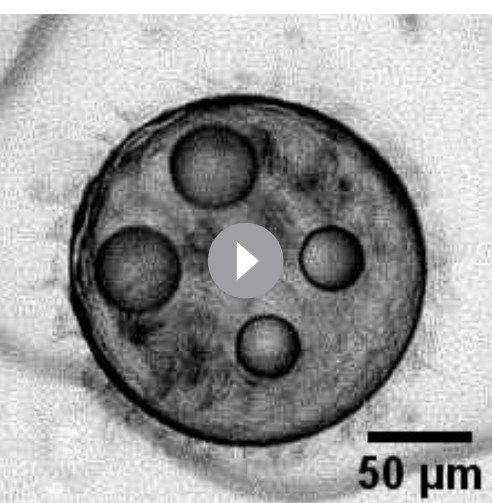

**Video 4.** Representative recording of a 2-day-old larva tethered to a glass cuvette and imaged from the anterior side to quantify the effect of pressure on the ciliary dynamics. Note how ciliary beating speeds up soon after the step change in pressure (998 mb).

https://elifesciences.org/articles/94306/figures#video4

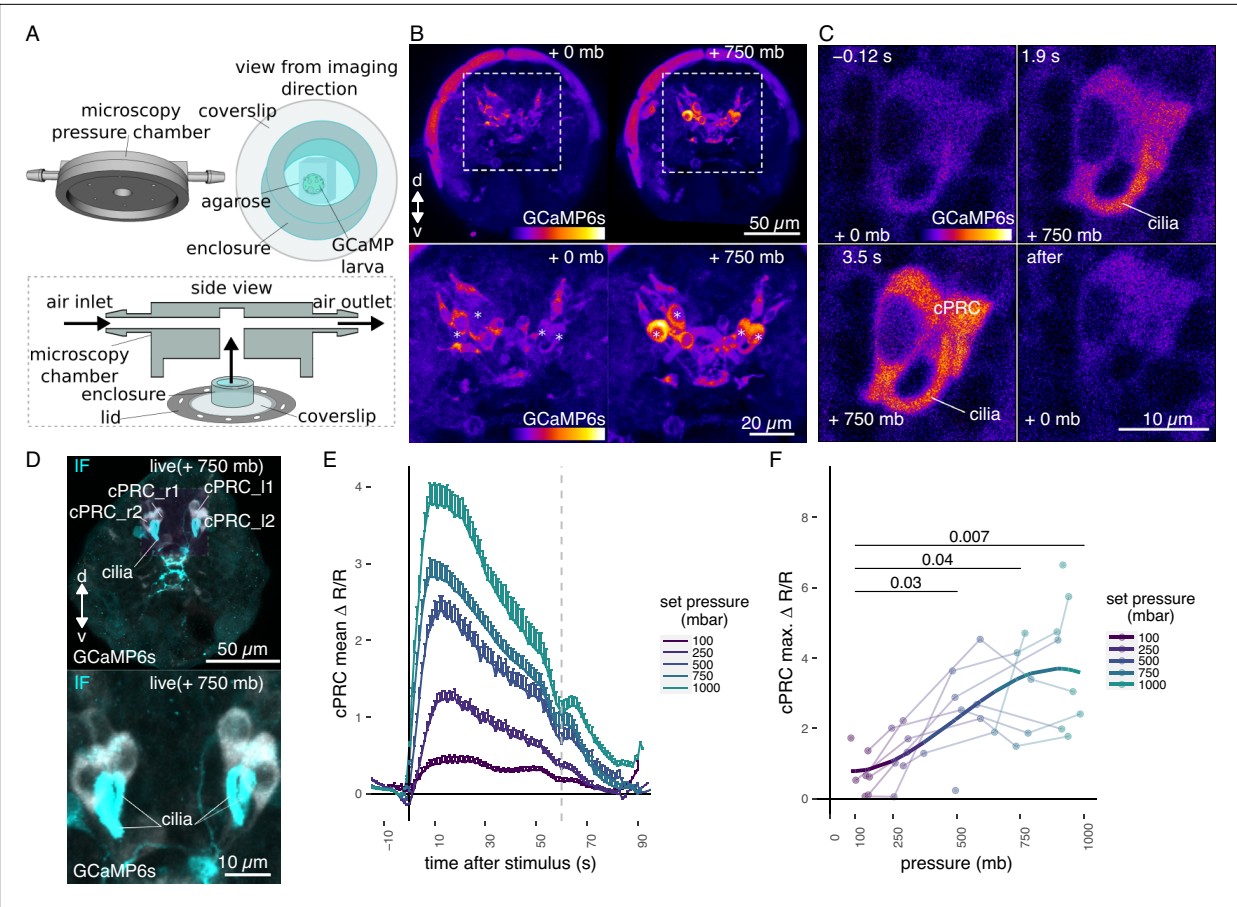

**Figure 2.** Ca²⁺ imaging of *Platynereis* larvae during pressure increases. (**A**) Ca²⁺ imaging preparation to analyse neuronal activation upon pressure stimulation. Left: side view of the three-dimensional (3D) model of the microscopy pressure chamber. Right: larvae embedded in agarose are placed on a round coverslip. An enclosure around the embedded larva serves to keep it under water. Bottom: the enclosed larva on the coverslip is inserted in the central hole of the chamber. A screwable lid secures the coverslip to the chamber and prevents air leaks. Pressure is increased with compressed air entering from one of the inlets. (**B**, top) Maximum intensity projection of a Z-stack acquired before (left) or during (right) the pressure stimulus of a 2-day-old larva expressing GCaMP6s. (**B**, bottom) Enlarged views of the corresponding regions highlighted with dashed squares in the top panels. Asterisks mark the position of cell nuclei of the four cells activated by pressure. (**C**) Still images of a ciliary photoreceptor cell (cPRC) acquired at different time points relative to increase in pressure (*t* = 0, *Video 5*). The time points are indicated on the upper left of each panel. Pressure level is also indicated. (**D**) Maximum intensity projection of a GCaMP6s Z-stack during raised pressure (white channel) and of a Z-stack of the same larva after immunofluorescence (IF) with NIT-GC2, a marker for cPRC cilia (*Jokura et al., 2023*), and for serotonin (cyan channel). Anterior view in **B–D**. (**E**) Mean Δ*R*/*R* in cPRC_l1 across different step increases in pressure as a function of time of stimulation. Dashed line at 60 s marks the end of stimulus. *N* = 8 larvae. Error bars show the standard error of the mean. (**F**) Max. Δ*R*/*R* in cPRC_l1 as a function of pressure level. Data points from the same larva are joined by lines. Regression line fitting the data is also shown. One-tailed unpaired *t*-test with Bonferroni correction testing for an increase in Max. Δ*R*/*R* with pressure. p-values <0.05 are shown. *Figure 2—source data 1* (**E, F**).

The online version of this article includes the following source data and figure supplement(s) for figure 2:

**Source data 1.** Raw intensity values of GCaMP6s (GC) and tdTomato (Tom) channels in cPRCs of 2-day-old larvae subjected to randomized step increases in pressure.

**Figure supplement 1.** Ca²⁺ imaging during pressure increase revealed cells activated during pressure.

**Figure supplement 2.** Ca²⁺ imaging in ciliary photoreceptor cells (cPRCs) and in SN^d1_unp.

imaging of neuronal activity with pressure increases (*Figure 2A*). We injected fertilized eggs with mRNA encoding the calcium indicator GCaMP6s (*Chen et al., 2013*)—an indirect reporter of neuronal activity—and embedded injected larvae in low-melting agarose. Mounted larvae were introduced into a custom-built microscopy chamber, where pressure could be increased using compressed air. To provide morphological landmarks and to correct for Z-shifts during imaging, we co-injected larvae with an mRNA encoding the membrane-tagged reporter palmitoylated tdTomato.

By imaging the entire larva before and during the pressure stimulus, we found a group of four cells in the dorsomedial brain that showed consistent increases in GCaMP6s fluorescence when pressure increased (*Figure 2B*; *Figure 2—figure supplement 1*). Time-lapse (TL) recordings of these cells revealed that they had prominent cilia, which become visible by the increase in GCaMP6s signal during pressure increase (*Figure 2C*, *Figure 2—figure supplement 1*, *Video 1*). The position, number, size, and morphology of these cells closely matched to that of the previously described brain ciliary photoreceptor cells (cPRCs) (*Arendt et al., 2004*; *Tsukamoto et al., 2017*; *Verasztó et al., 2018*). Immunostaining of the same larvae that were used for Ca²⁺ imaging with an antibody raised against NIT-GC2, a marker of cPRC cilia(*Jokura et al., 2023*), followed by image registration directly confirmed that the four cells activated under pressure were the cPRCs (*Figure 2D*).

To characterize the response of cPRCs to pressure, we applied a randomized set of pressure increases (*Figure 2—figure supplement 2A*) to 2-day-old larvae expressing GCaMP6s while recording fluorescence changes in the four cPRCs. All four cPRCs responded to the pressure levels tested (*Figure 2E*; *Figure 2—figure supplement 2B, C*). The increase in GCaMP6s signal was observed in both the cell body and in the cilia of the cPRCs (*Figure 2C*, *Video 5*). This response—like that observed at the behavioural level—was graded and increased proportionally to the pressure change (*Figure 2E, F*; *Figure 2—figure supplement 2B, C, E*). The difference in the response between pressure levels was statistically significant for some of the cPRCs (*Figure 2F*; *Figure 2—figure supplement 2C*). The calcium signal also decreased rapidly after stimulus onset. Therefore, cPRCs may be able to directly encode the intensity of the stimulation in their activity, reflected in their internal Ca²⁺ levels, and adapt to pressure levels. Their unique sensory morphology and pressure-induced Ca²⁺ dynamics make the cPRCs candidate pressure receptors.

An additional unpaired sensory cell on the dorsal side was also activated in some of the trials (*Figure 2—figure supplement 1*, green asterisk). We refer to this cell here as SN^d1_unp (by position and morphology it corresponds to the neurosecretory cell SN^YFa+; *Williams et al., 2017*). At 750 mb, this cell responded by a transient but robust increase at stimulation onset, but $\Delta R/R$ dropped to basal values before the end of the stimulus, unlike the cPRC response (*Figure 2—figure supplement 2D*). SN^d1_unp (SN^YFa+) may also contribute to the pressure response, although it is less sensitive than cPRCs and has very few synapses (*Williams et al., 2017*).

Another indirect observation that is consistent with cPRCs being the primary pressure receptors is that 1-day-old larvae that lack differentiated cPRCs (*Fischer et al., 2010*) do not respond to pressure (*Figure 1—figure supplement 2A*).

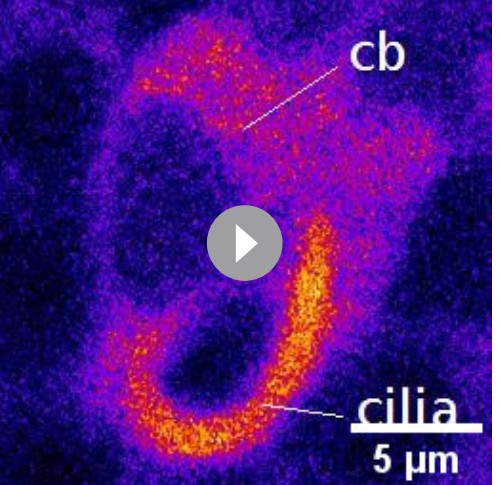

**Video 5.** Time-lapse confocal microscopy recording showing the change in GCaMP6s fluorescence (fire colour scale) in a ciliary photoreceptor cell (cPRC) in response to pressure increase (750 mb). The increase in GCaMP6s signal is noticeable both in the cell body (cb) and in the cilia. Some frames are shifted in *XY* due to sample movement.

https://elifesciences.org/articles/94306/figures#video5

## c-opsin-1 mutants have a reduced pressure response

cPRCs express ciliary-opsin-1 (c-ops-1), which forms a UV-absorbing photopigment (*Arendt et al., 2004*; *Tsukamoto et al., 2017*; *Verasztó et al., 2018*). Knocking out the *c-ops-1* gene abolishes a UV-avoidance response in *Platynereis* larvae (*Verasztó et al., 2018*). As cPRCs respond to pressure increases, we tested whether *c-ops-1* knockout mutants (*c-ops-1^Δ8/Δ8*) also showed a defect in the pressure response.

A range of step increases in pressure were applied to single batches of either wild-type (WT) or *c-ops-1^Δ8/Δ8* 3-day-old larvae (*Figure 3—figure supplement 1A*). The assays were carried out in a smaller pressure vessel (height: ~4 cm), to allow consistent imaging of the fewer mutant larvae available. The swimming speed of *c-ops-1^Δ8/Δ8* mutant larvae was not significantly different from WT larvae (p = 0.118, unpaired Wilcoxon test for lower speed in mutants; *Figure 3—figure supplement 1B*). *c-ops-1^Δ8/Δ8*

larvae responded in a graded manner to increases in pressure by upward swimming (*Figure 3A*; *Figure 3—figure supplement 1C*). However, their response was weaker than the response of age-matched WT larvae (*Figure 3—figure supplement 1C*). An ANOVA comparison showed that a model considering the genotype better explained the data than a model without this variable, for either a simple or a polynomial linear regression model (p-values = $9.98e^{-07}$ and $9.92e^{-06}$, respectively). Ciliary beating prior to pressure increase was not significantly different between *c-ops-1$^{\Delta 8/\Delta 8}$* and WT larvae (p = 0.835, unpaired Wilcoxon test for lower CBF in mutants; *Figure 3—figure supplement 1D*). Upon pressure increases, CBF in *c-ops-1$^{\Delta 8/\Delta 8}$* larvae showed a significant increase to the three highest pressure levels tested (237.5 mb p = 0.014, 556 mb p = 9.7E−05, 988 mb p = 6.35E−05; one-tailed paired *t*-test with Bonferroni correction testing for an increase in CBF; *Figure 3B*). *c-ops-1$^{\Delta 8/\Delta 8}$* larvae also showed significant increases in max. Δ%CBF as the pressure stimulus was increased (*Figure 3—figure supplement 1E*; compare to the WT data in *Figure 3B*), with no significant difference in the increase to WT larvae (*Figure 3—figure supplement 1F*). In summary, *c-ops-1$^{\Delta 8/\Delta 8}$* larvae can still respond to changes in pressure in a graded manner, but the response is weaker than in WT larvae, both at the population and at the single-larva levels. This indicates that c-opsin-1 is not directly required for the pressure response, but its absence leads to a weakened response to pressure.

## c-opsin-1 mutants have defects in cPRC cilia

The reduced response of *c-ops-1$^{\Delta 8/\Delta 8}$* larvae to pressure may stem from morphological defects of the cPRC sensory cilia in these mutants. We used stainings with acetylated tubulin and NIT-GC2, an antibody specifically marking cPRC cilia (see *Figure 2D*) to measure the volume of the ciliary compartment in WT and *c-ops-1$^{\Delta 8/\Delta 8}$* larvae (*Figure 3C*; *Figure 3—figure supplement 1G*). Volumetric imaging in 2-day-old larvae stained with these antibodies revealed that the cPRC ciliary compartment in *c-ops-1$^{\Delta 8/\Delta 8}$* larvae was significantly smaller than in age-matched WT larvae (p = 0.044 and 0.018 for left and right ciliary compartments, one-tailed unpaired *t*-test with Bonferroni correction; *Figure 3D*). In some cases, cPRC ciliary compartments were drastically reduced, albeit never completely absent in all four cells (*Figure 3—figure supplement 1F*). A reduced ciliary compartment may underlie the weaker responses of *c-ops-1$^{\Delta 8/\Delta 8}$* larvae to pressure.

To further investigate the morphological defects of cPRC sensory cilia in *c-ops-1$^{\Delta 8/\Delta 8}$* larvae, we reconstructed the cPRC ciliary structure of a mutant larva using volume electron microscopy. We compared this reconstruction to a volume EM dataset of cPRC cilia from a 3-day-old WT larva previously reported (*Verasztó et al., 2018*). The two WT cPRCs reconstructed have 14 to 15 ramified cilia tightly wrapped on themselves (*Figure 3E*, top row, *Video 6*). Branching occurs close to the basal body, soon after cilia protrude from the cell body. Most branches inherit an individual microtubule doublet. The 15 to 17 cilia of the two cPRCs reconstructed in a *c-ops-1$^{\Delta 8/\Delta 8}$* larva revealed a more sparsely packed structure (*Figure 3E*, bottom row; *Figure 3—figure supplement 1A*, *Video 7*). Mutant cilia are not significantly shorter than WT cPRC cilia (p = 0.076, Wilcoxon test for longer WT cilia, *Figure 3F*). However, we noticed when comparing cross-sections of each genotype that individual branches of mutant cPRC cilia often contained more than one microtubule doublet (arrowheads in *Figure 3E*; *Figure 3—figure supplement 2A*). This suggests that cPRC cilia of *c-ops-1$^{\Delta 8/\Delta 8}$* larvae have alterations in branching morphology. We indeed found that the terminal branches of cilia in the mutant are significantly shorter (p = 6.29E−05 Wilcoxon test for shorter branches in the mutant, *Figure 3G*, right-most plot), while internal branches are longer than those in the WT (p = 1.26E−04; Wilcoxon test for longer branches in the mutant, *Figure 3G*, middle plot). Basal branches also tended to be larger, but no statistically significance can be concluded (p = 0.06, Wilcoxon test for longer branches in the mutant, *Figure 3G*, left-most plot). This result supports the former observation that branching occurs more distally to the basal body in *c-ops-1$^{\Delta 8/\Delta 8}$* larvae. Longer internal branches in mutant cPRC cilia would also explain the presence of ciliary profiles with more than one microtubule doublet (*Figure 3—figure supplement 2A*).

Overall, our physiological and genetic analyses suggest that the brain cPRCs act as graded and fast-adapting pressure receptors. In *c-ops-1$^{\Delta 8/\Delta 8}$* larvae, the ciliary compartment is smaller and shows morphological defects, revealing a genetic requirement for c-opsin-1 in the establishment of the sensory compartment and supporting the idea that the cPRC ciliary compartment is the site of pressure transduction.

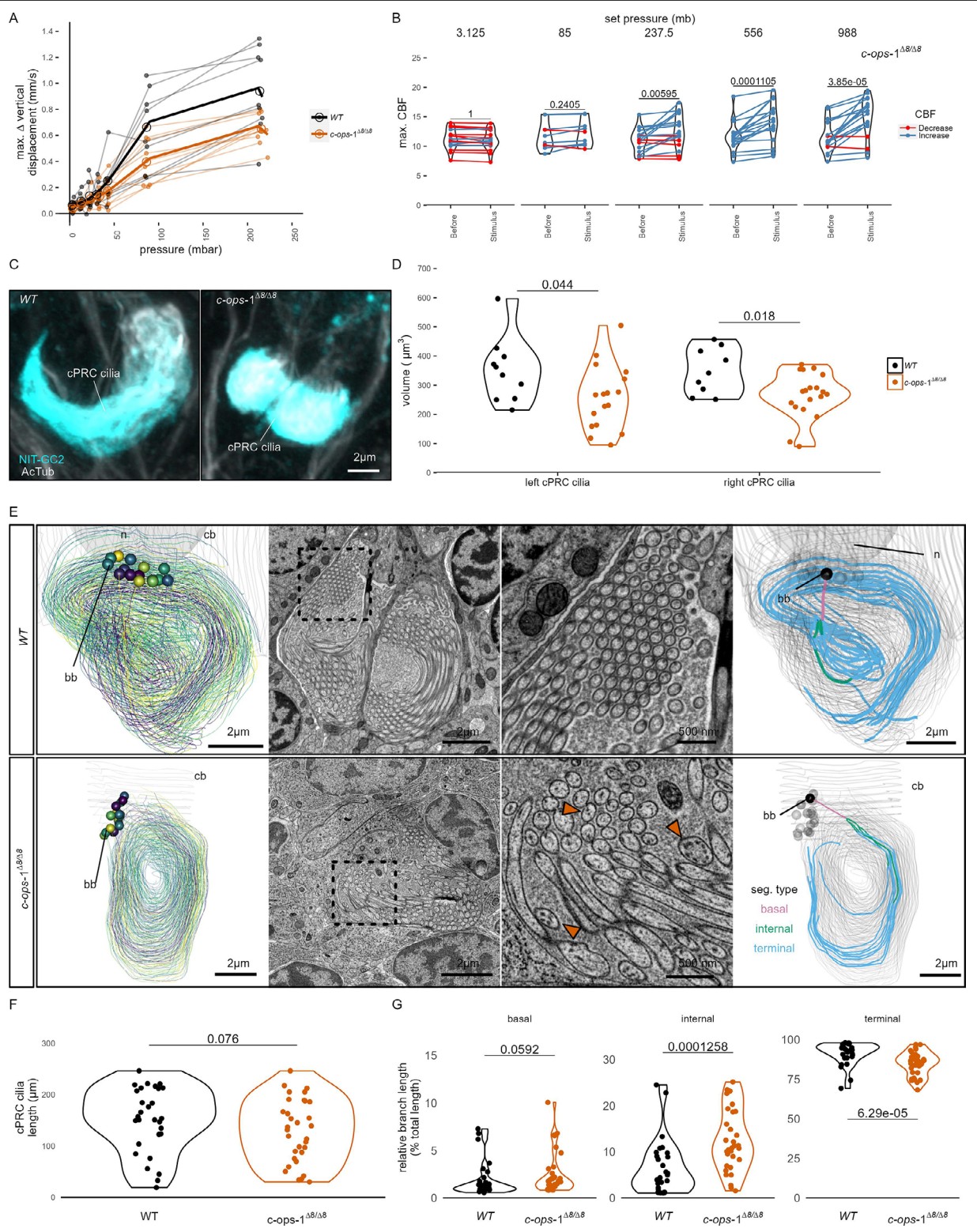

**Figure 3.** *c-ops-1*$^{\Delta 8/\Delta 8}$ larvae show a weaker response to pressure and have structural defects in ciliary photoreceptor cell (cPRC) cilia. (**A**) Maximum change in vertical displacement of wild-type (WT) and *c-ops-1*$^{\Delta 8/\Delta 8}$ 3-day-old larvae for each pressure step-change tested. Data points from the same batch are joined by lines. Larger open circles indicate the mean value. Thicker solid lines show the regression model predictions. *N* = 7–9 (*c-ops-1*$^{\Delta 8/\Delta 8}$), *N* = 8–10 (WT) batches. (**B**) Max. ciliary beat frequency (CBF) that individual *c-ops-1*$^{\Delta 8/\Delta 8}$ larvae reached in the 30 s prior (Before), or during the first 30 s of the indicated increase in pressure (Stimulus). Data points for the same larva are joined by lines. One-tailed paired *t*-test with Bonferroni correction

*Figure 3 continued on next page*

*Figure 3 continued*

testing for an increase in CBF; all p-values are shown. *N* = 10–17 larvae. (**C, D**) cPRC ciliary volume measured for each pair of cPRC cilia on the left and right body sides in 2-day-old WT and *c-ops-1*[Δ8/Δ8] larvae. Volumes were measured using the signal of the NIT-GC2 antibody (α-NIT-GC2). (**C**) Maximum intensity projections of IF stainings used for quantifying cPRC cilary volume. (**D**) cPRC ciliary volume distribution sorted by genotype and body side. One-tailed unpaired *t*-test with Bonferroni correction testing for a decrease in ciliary volume in mutant larvae. p-values <0.05 are shown. *N* = 9–10 (WT), 17–19 (*c-ops-1*[Δ8/Δ8]) larvae. (**E**) Morphology of cPRC cilia in 3-day-old WT (top row) and *c-ops-1*[Δ8/Δ8] (bottom row) larvae reconstructed by serial-section electron microscopy (ssEM). Reconstructions of cPRC cilia are shown in the left-most panels. A representative micrograph of the ssEM data used for the reconstructions is shown in the adjacent panels. Dashed squares in these images mark the regions shown in the enlarged views to the right. Orange arrowheads point to cilia with more than one microtubule dublets. In the right-most panels, the branches of single cPRC cilia are coloured by its position relative to the basal body (bb): basal, internal, or terminal branches. The remaining cPRC cilia are coloured in grey. cb: cell body; n: nucleus. (**F**) Length of cPRC cilia of WT (N = 29 cilia) and *c-ops-1*[Δ8/Δ8] (N = 32 cilia) larvae measured from ssEM volume reconstructions. Unpaired Wilcoxon test for larger branches in the WT larva: p = 0.076. (**G**) Length distribution of basal, internal and terminal branches for each genotype shown as a percentage of the total ciliary arbor length. One-tailed unpaired Wilcoxon test with Bonferroni correction for larger basal, internal, and terminal branches in the *c-ops-1*[Δ8/Δ8] mutant: p = 0.0592, p = 1.26E-4, p = 6.29E-5, respectively. *Figure 3—source data 1* (**A**), *Figure 1—source data 4* (**B**), *Figure 3—source data 2* (**D**).

The online version of this article includes the following source data and figure supplement(s) for figure 3:

**Source data 1.** Swimming metrics of individual batches of 3-day-old wildtype or *c-opsin-1* mutant larvae subjected to randomized step increases in pressure.

**Source data 2.** Volume of ciliated structure of individual cPRCs of wildtype or *c-opsin-1* mutant 2-day-old larvae measured by quantifying the NIT-GC2 antibody signal.

**Figure supplement 1.** Response of 3-day-old *c-ops-1*[Δ8/Δ8] larvae to pressure.

**Figure supplement 2.** Morphology of ciliary photoreceptor cell (cPRC) cilia in 3-day-old *c-ops-1*[Δ8/Δ8] larvae.

## Synaptic transmission from serotonergic ciliomotor neurons mediates pressure-induced increases in ciliary beating

The complete synaptic wiring diagram of the cPRCs was previously reported from an electron microscopy volume of a 3-day-old larva (*Figure 4A, B*; *Verasztó et al., 2018*). The shortest neuronal path from cPRCs to the prototroch involves a feed-forward loop from the cPRCs to two types of postsynaptic interneurons: the INNOS and the INRGWa cells, expressing respectively Nitric Oxide Synthase (NOS) or the neuropeptide RGWamide (*Figure 4B*). INRGWa cells in turn synapse on a pair of head serotonergic ciliomotor neurons (Ser-h1). Ser-h1 cells directly innervate the prototroch ciliary band and synapse on the MC head cholinergic ciliomotor neuron (*Figure 4B*). Ser-h1 cells are thought to promote ciliary beating by directly releasing serotonin onto the ciliary band cells and indirectly by inhibiting the cholinergic MC neuron that is required to arrest ciliary beating (*Verasztó et al., 2017*).

To directly test the involvement of the Ser-h1 cells in the pressure response, we used a genetic strategy to inhibit synaptic release from serotonergic neurons. We used transient transgenesis to drive the expression of the synaptic inhibitor tetanus-toxin light chain (TeTxLC) (*Sweeney et al., 1995*) under the promoter of *tryptophan hydroxylase* (*TPH*), a marker of serotonergic neurons (*Verasztó et al., 2017*). The *TPH* promoter labels the head Ser-h1 and other serotonergic cells including the Ser-tr1 trunk ciliomotor neurons. Of the cells labelled with this promoter, only Ser-h1 is postsynaptic to cPRCs and provides strong innervation to the prototroch cells and the MC cell (*Verasztó et al., 2017*).

The construct (*Figure 4C*) drives the expression of both TeTxLC and a hemagglutinin (HA) tagged palmitoylated tdTomato reporter, separated by P2A, a self-cleaving peptide previously used in *Platynereis* (*Bezares-Calderón et al., 2018*). The mosaic expression of this construct resulted in the labelling of different subsets of serotonergic neurons. We selected animals showing labelling in Ser-h1 (most larvae were also labelled in Ser-tr1 and other unidentified serotonergic cells). We confirmed the expression of the transgene in Ser-h1 by immunostaining against the HA-tag of Palmi-tdTomato (*Figure 4D*). Larvae expressing TeTxLC in Ser-h1 showed an increase in CBF only at the highest pressure used (988 mb p = 0.004; one-tailed paired *t*-test with Bonferroni correction testing for an increase in CBF; *Figure 4E*, bottom row). Larvae injected with a control plasmid expressing only Palmi-tdTomato but not TeTxLC in Ser-h1 (*Figure 4C*) showed a significant increase in CBF at the three highest pressure levels applied (237.5 mb p = 0.023, 556 mb p = 0.009, 988 mb p = 0.004; one-tailed paired *t*-test with Bonferroni correction testing for an increase in CBF; *Figure 4E*, top row; *Figure 4—figure supplement 1A*). The metric max. Δ%CBF was not significantly different between control and TeTxLC-injected larvae (*Figure 4—figure supplement 1B*). These results indicate that TeTxLC-mediated inhibition of Ser-h-1—albeit incomplete and in most cases limited to one of the two cells—leads to a

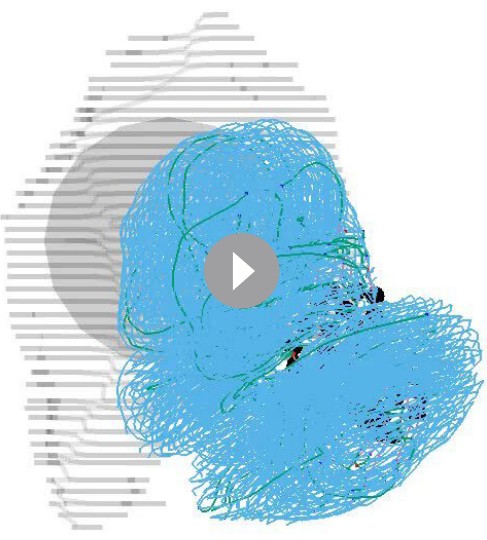

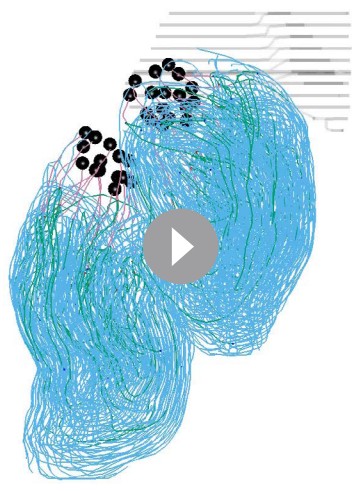

**Video 6.** 360° view of electron microscopy reconstruction of sensory cilia from one pair of ciliary photoreceptor cells in a wild-type 3-day-old larva. The branches of each cilium are coloured according to their position along the cilium: basal (pink), intermediate (green), or light blue (terminal). Basal bodies are shown as black spheres. The cell and nucleus of one of the cells are outlined (previously reconstructed in *Verasztó et al., 2018*).
https://elifesciences.org/articles/94306/figures#video6

**Video 7.** 360° view of electron microscopy reconstruction of sensory cilia from one pair of ciliary photoreceptor cells in a *c-ops-1*$^{\Delta 8/\Delta 8}$ 3-day-old larva. The branches of each cilium are coloured according to their position along the cilium: basal (pink), intermediate (green), or light blue (terminal). Basal bodies are shown as black spheres. The part of the cell body imaged in the volume is outlined.
https://elifesciences.org/articles/94306/figures#video7

noticeable dampening of the pressure response at the level of the ciliary band. This suggests that synaptic release from serotonergic neurons is required to increase ciliary beating upon pressure increase. Our data support the model that Ser-h1 neurons, and no other serotonergic cells, are specifically required for this response, because this cell type was labelled in all animals tested and these cells directly innervate the prototroch.

## Discussion

This work provides insights into the neuronal mechanisms of pressure sensation and response in a marine planktonic larva. Our findings suggest that increases in pressure, either due to the larva's own actions (sinking or diving) or to downwelling currents, lead to the activation of the sensory cPRCs (*Figure 5*). The activation is proportional to the magnitude of pressure change and leads to the activation of the downstream circuit that converges onto the Ser-h1 neurons and ultimately leads to increased ciliary beating. The Ser-h1 cells could secrete serotonin (5-HT) onto the ciliary band, which from pharmacological assays is known to increase CBF (*Verasztó et al., 2017*). Ser-h1 neurons may simultaneously inhibit the cholinergic ciliomotoneuron MC neuron (*Verasztó et al., 2017*), thereby preventing ciliary arrests while the larva tries to compensate for the increased pressure. Upon a decrease in pressure, 3-day-old (but not 2-day-old) larvae also show an off-response characterized by downward swimming. We have not analysed in detail the neuronal mechanisms of this response but it may depend on an inverted activation of the cPRC circuit, as happens during UV avoidance (*Jokura et al., 2023*). Pressure 'on' and pressure 'off' thus induce behavioural responses with opposite sign such that larvae move to compensate for the pressure change. The response is directional along the pressure gradient (even if larvae do not detect the gradient but temporal changes) and we refer to it as barotaxis.

Our proposed depth-retention model complements a previously reported spectral depth-gauge mechanism in the *Platynereis* larva based on its ability to respond to UV and green light (*Verasztó et al., 2018*). As previously hypothesized for other planktonic larvae (*Sulkin, 1984*), pressure sensing in *Platynereis* may operate through negative feedback to retain a particular depth, while light as a more variable stimulus could drive intensity and wavelength-dependent changes in the vertical

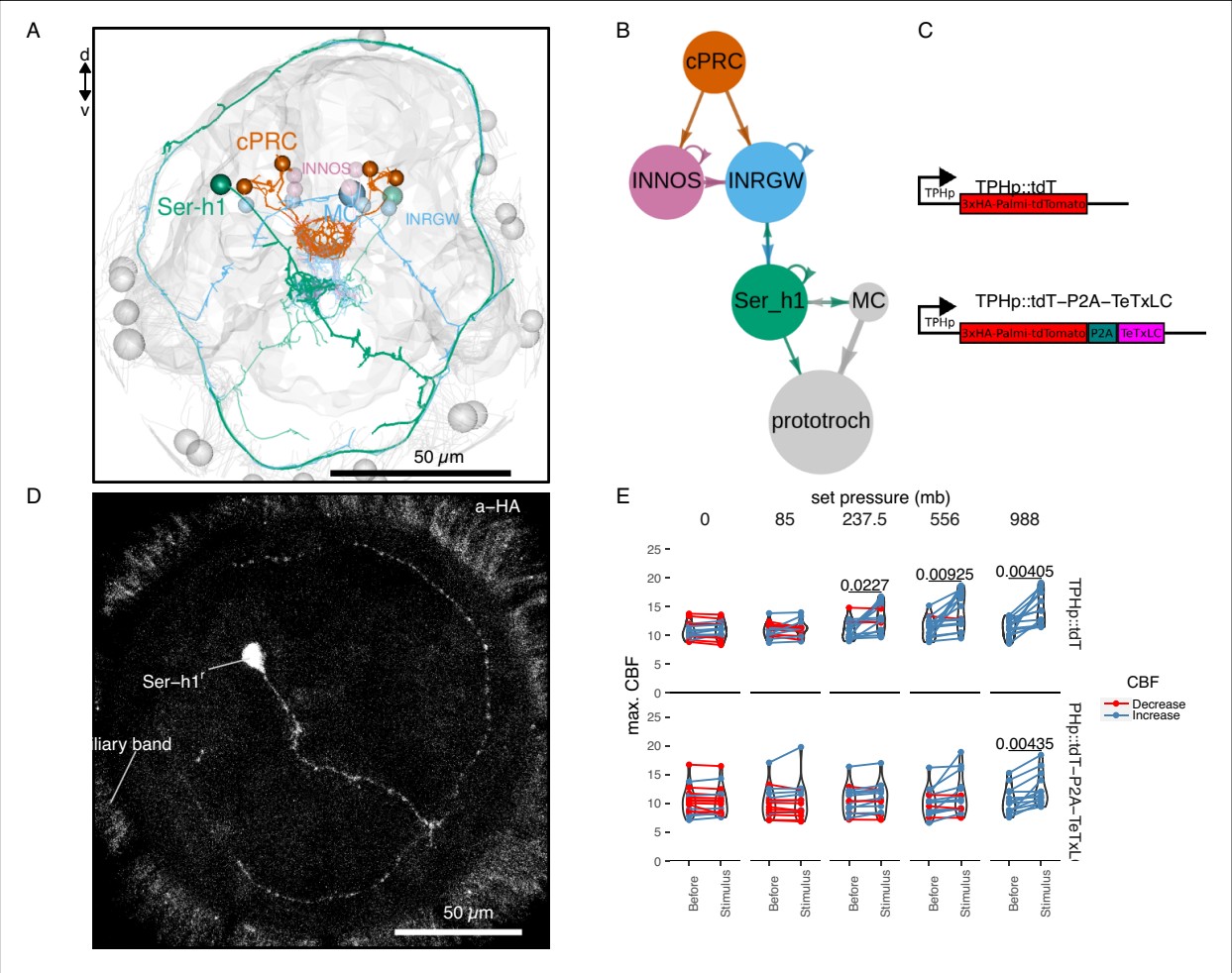

**Figure 4.** TeTxLC expression in Ser-h1 neurons inhibits ciliary beat frequency (CBF) increase during pressure. Volume EM reconstruction (**A**) of the cells in the ciliary photoreceptor cell (cPRC) synaptic circuit (**B**). cPRCs synapse in the middle of the brain on both INNOS and INRGWa neurons. INRGWa cells synapse on Ser-h1 neurons, which directly innervate the ciliary band and also target the MC cell. Only the cells in the shortest path to the prototroch are included. (**C**) Schematic of the gene constructs used to test the role of Ser-h1 in the pressure response. The TPHp::tdT construct drives expression of the reporter protein Palmi-tdTomato in Ser-h1 and other serotonergic neurons. It was used as a control. The TPHp::tdT-P2A-TeTxLC construct expresses Palmi-tdTomato and the synaptic blocker TeTxLC as a fusion that gets post-translationally self-cleaved by the P2A peptide. (**D**) Maximum intensity projections of a larva injected with the TPHp::tdT-P2A-TeTxLC construct and stained with α-HA. Ser-h1 was labelled in this animal. Anterior view. (**E**) Max. CBF that single larvae reached in the 30 s prior (Before), or during the first 30 s of the indicated increase in pressure (Stimulus). Data points for the same larva are joined by lines. Larvae were injected either with the control plasmid TPHp::tdT (top row, N = 14–16 larvae), or with the TPHp::tdT-P2A-TeTxLC plasmid (bottom row, N = 13–16 larvae). One-tailed paired *t*-test with Bonferroni correction testing for an increase in CBF; p-values <0.05 are shown. *Figure 4—source data 1* (**E**).

The online version of this article includes the following source data and figure supplement(s) for figure 4:

**Source data 1.** Ciliary band dynamics of individual 2-day-old wildtype larvae injected with pLB316 or with pLB253 plasmid constructs and subjected to randomized step increases in pressure.

**Figure supplement 1.** Effect of TeTxLC-mediated inhibition of serotonergic ciliomotor neurons on ciliary beat frequency (CBF) during pressure stimulation.

position of larvae. In future, it will be interesting to explore how responses tp pressure and to directional (*Randel et al., 2014*) and non-directional light cues (*Verasztó et al., 2018*) across wavelengths and intensities interact to guide larval swimming.

Our unexpected finding that ciliary photoreceptors, previously shown to be sensitive to UV and green light (*Verasztó et al., 2018*), are also activated by pressure increase suggests that the integration of light and pressure could begin at the sensory level in a single-cell type. UV light activates the cPRCs to induce downward swimming while pressure increases induce upward swimming.

The calcium dynamics of cPRCs depends on the stimulus applied and may therefore underlie the mechanism by which the cells decode and transmit a sensory signal to the downstream circuit. UV induces a transient increase and subsequent drop in $Ca^{2+}$ levels followed by a sharp NO-dependent increase that persists even after UV stimulation ends (*Jokura et al., 2023*). In contrast, pressure induces a transient increase in $Ca^{2+}$ levels that terminates with the stimulus. The different activation dynamics may lead to the release of different neurotransmitter and neuromodulator cocktails cPRCs are cholinergic, GABAergic, adrenergic, and peptidergic (*Jokura et al., 2023*; *Randel et al., 2014*; *Tessmar-Raible et al., 2007*; *Williams et al., 2017*) and different activation patterns in the post-synaptic cells. Similar integration can occur in *Drosophila* larvae, where UV/blue light and noxious mechanical stimuli are detected by the same sensory neuron that together with other cells converges onto a circuit processing multisensory stimuli (*Imambocus et al., 2022*). In *Platynereis*, other sensory cells (e.g. the SN<sup>d1_unp</sup> cells; *Figure 2—figure supplement 1*) may contribute to distinguishing the nature of the stimulus. How UV and pressure signals are integrated by the cPRC and how other light responses such as phototaxis interact with pressure responses remain exciting avenues for future research.

The cellular and molecular mechanisms by which cPRCs sense and transduce changes in hydrostatic pressure deserve further enquiry. The mechanism may involve the differential compression of microtubules along each ciliary branch (*Li et al., 2022*; *Nasrin et al., 2021*), or differential displacement of fluid inside the cilium (*Bell, 2008*). The cPRCs have a unique multiciliated structure and are embedded in a protected environment different from fluid-exposed cilia such as hydrodynamic mechanosensory cilia (*Bezares-Calderón et al., 2018*). These features are hard to reconcile with current models of ciliary mechanosensation (e.g. see review by *R Ferreira et al., 2019*). Instead, the mechanism of pressure sensation at work in cPRCs may share more similarities with non-neuronal cells detecting pressure, such as chondrocytes (*Pattappa et al., 2019*) and trabecular meshwork cells (*Luo et al., 2014*).

The molecular mechanisms of pressure detection remain unclear. Components of the phototransduction cascade may be involved in pressure sensation. Our results indicate that the ciliary opsin required for detecting UV light is not essential for pressure sensation. This molecule rather indirectly affects the ability of cPRCs to sense pressure by contributing to the development or maintenance of a ramified ciliary sensory structure. The structural role of opsins for shaping ciliary cell morphology has also been reported in other photoreceptor and mechanoreceptor cells (*Lem et al., 1999*; *Zanini et al., 2018*). The direct transducer of pressure may be a Transient Receptor Potential (TRP) channel. TRP channels can signal downstream of opsins in phototransduction cascades and have been postulated as the ultimate integrators of sensory stimuli (*Liu and Montell, 2015*). Mechanosensitivity has also recently been reported for vertebrate rods (*Bocchero et al., 2020*), including the activation of these ciliary photoreceptors by pressure (*Pang et al., 2021*). The cellular and molecular mechanisms behind this sensitivity are still unclear, but TRP or Piezo channels were suggested to be involved (*Bocchero et al., 2020*).

The mechanism by which cPRCs detect pressure in *Platynereis* may also characterize the few other cases of pressure sensors based on ciliated sensory cells (the crab statocyst, and the dog hair cells) (*Fraser and Macdonald, 1994*; *Fraser and Shelmerdine, 2002*). Multiciliated pressure receptors may be widespread in zooplankton, as cells with a morphology similar to the cPRCs have been observed in species across the animal phylogeny (*Hernandez-Nicaise, 1984.*; *Baatrup, 1982*; *Eakin and Kuda, 1970*), reviewed in *Bezares-Calderón et al., 2020*. Some of these cells have long been hypothesized to function as pressure sensors. To our knowledge, this study provides the first functional evidence for the role of multiciliated sensory cells in pressure sensation.

cPRCs are part of an ancient neurosecretory centre with putative roles in circadian regulation of locomotion (*Tessmar-Raible et al., 2007*; *Tosches et al., 2014*; *Williams et al., 2017*). Besides light, pressure can also regularly change as a result of dial vertial migration in planktonic stages or tides in settled benthic stages. One of the original components in the first neurosecretory centres may thus have been multimodal sensory cells keeping track of and integrating changes in periodic cues such as light and pressure that allowed to coordinate physiology and behaviour with the natural rhythms.

# Materials and methods

## Key resources table

| Reagent type (species) or resource | Designation | Source or reference | Identifiers | Additional information |
|---|---|---|---|---|
| Strain, strain background (*Platynereis dumerilii*) | Wild-type | Marine Invertebrate Culture Unit, University of Exeter | NCBITaxon:6359 | |
| Strain, strain background (*P. dumerilii*) | c-opsin1Δ8/Δ8 knockout | *Verasztó et al., 2018* | NCBITaxon:6359 | Knockout generated by TALEN-induced gene editing |
| Transfected construct (mRNA) | GCaMP6s mRNA | *Randel et al., 2014* | | 1 mg/ml |
| Transfected construct (mRNA) | 3xHA-Palmi-tdTomato mRNA | This paper | | <0.2 ng/µl |
| Antibody | Monoclonal Anti-Tubulin, Acetylated antibody produced in mouse | Sigma-Aldrich | Cat# T6793, RRID:AB_477585 | (1:250) |
| Antibody | Polyclonal NIT-GC2 antibody, produced in rabbit | *Jokura et al., 2023* | aNIT-GC2 | 5 mg/ml |
| Antibody | anti-5-HT, produced in rabbit | ImmunoStar | Cat# 20080, RRID:AB_572263 | 2 mg/ml |
| Antibody | Monoclonal Anti-HA antibody produced in mouse | Cell Signaling Technology | HA-Tag (6E2), Cat# 2367, RRID:AB_10691311 | (1:250) |
| Antibody | Monoclonal Anti-HA antibody produced in rabbit | Cell Signaling Technology | HA-Tag (C29F4) Cat #3724, RRID:AB_1549585 | (1:250) |
| Antibody | Goat anti-Rabbit IgG (H+L) Cross-Adsorbed Secondary Antibody, Alexa Fluor 488 produced in rabbit | Thermo Fisher Scientific | Cat# A-11008, RRID:AB_143165 | (1:250) |
| Antibody | F(ab')2-Goat anti-Mouse IgG (H+L) Cross-Adsorbed Secondary Antibody, Alexa Fluor 546 | Thermo Fisher Scientific | Cat# A-11018, RRID:AB_2534085 | (1:250) |
| Commercial assay or kit | mMESSAGEmMACHINE T7ULTRA Transcription Kit | Ambion, Thermo Fisher Scientific | Cat# AM1345 | |
| Recombinant DNA reagent | pUC57-TPHp3xHA-Palmi-tdTomato-P2A-TeTxLC (plasmid) | This paper | pLB316 | Injected at 250 ng/µl in water |
| Recombinant DNA reagent | pUC57-TPHp3xHA-Palmi-tdTomato (plasmid) | *Verasztó et al., 2017* | pLB253 | Injected at 250 ng/µl in water |
| Recombinant DNA reagent | pUC57-T7-RPP2-3xHA-Palmi-tdTomato (plasmid) | This paper | pLB260 | Injected at 250 ng/µl in water |
| Recombinant DNA reagent | pUC57-T7-RPP2-GCaMP6s (plasmid) | *Randel et al., 2014* | pLB112 | Injected at 250 ng/µl in water |
| Recombinant DNA reagent | pGEMTEZ-TeTxLC | *Yu et al., 2004* | Addgene plasmid # 32640, RRID:Addgene_32640 | Richard Axel & Joseph Gogos & C. Ron Yu |
| Chemical compound, drug | Ficoll PM70 | Sigma-Aldrich | Cat# F2878 | −20% |
| Chemical compound, drug | Formaldehyde Aqueous Solution (Paraformaldehyde Aqueous Solution) EM Grade | Electron Microscopy Sciences | Cat# 15710 | −4% |
| Chemical compound, drug | Low-melting agarose | Hampton Research | LM AgaroseTM, Cat# HR8-092 | (2.5–3%) |
| Software, algorithm | Fiji | NIH | RRID:SCR_002285 | |
| Software, algorithm | CATMAID | *Saalfeld et al., 2009* | RRID:SCR_006278 | |
| Other | Wormglu | GluStitch Inc | | |

## Animal culture

*P. dumerilii* worms were cultured in a laboratory facility following established protocols. Larvae were raised in 0.22-µm-filtered artificial sea water (fASW, Tropic Marin) at 37 ppm and kept at 18°C in a 16/8-hr light–dark cycle. The genotype of each batch made from crosses of *c-ops-1* KO mutant (*c-ops-1Δ8/Δ8*) worms was confirmed using the primer set reported in *Verasztó et al., 2018*.

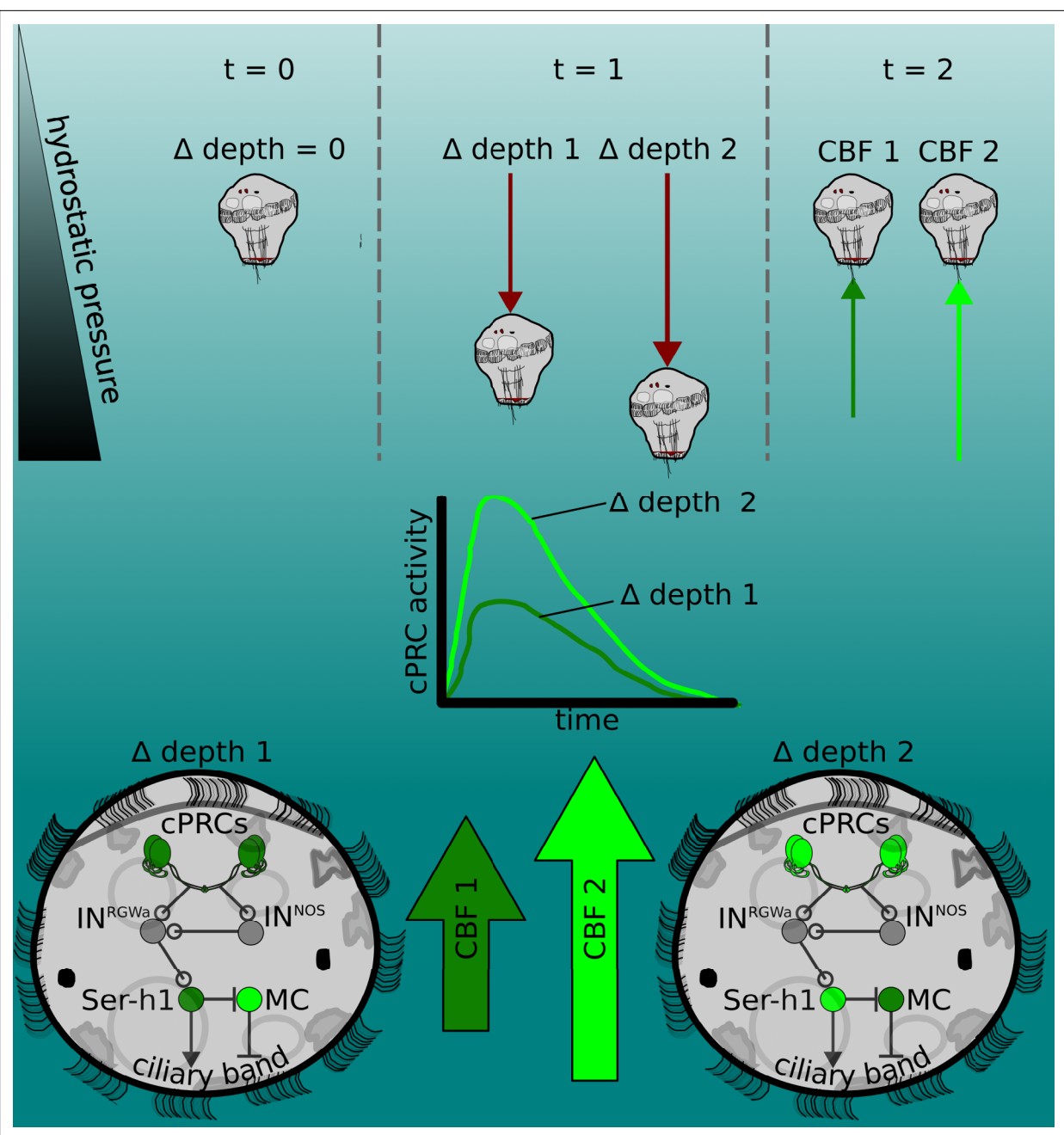

**Figure 5.** The pressure response as a depth-retention mechanism mediated by the ciliary photoreceptor-cell circuit. (Top) *Platynereis* larvae maintain their position in the water column by controlling the beating dynamics of the ciliary band (*t* = 0, no net change in depth, Δ depth = 0). A sufficiently rapid increase in depth (at *t* = 1) caused by intrinsic or extrinsic factors would lead to an increase in hydrostatic pressure. The change in depth relative to the previous state (Δ depth 1 or Δ depth 2) will be perceived by the larva, which will try to counteract this change by increasing ciliary beat frequency (CBF) of the prototroch, leading to upward swimming (at *t* = 2) until the pressure returns to the level previously experienced. A smaller change in pressure (Δ depth 1) will lead to a smaller increase in CBF (CBF 1, light green arrow) than a larger change (CBF 2). (Middle) Changes in pressure are sensed by the ciliary photoreceptor cells (cPRCs). These sensory cells are activated in a graded manner according to the change in pressure. (Bottom) cPRCs signal via a postsynaptic circuit including the INRGWa and INNOS interneurons to the Ser-h1 neurons. Activation of these serotonergic neurons is required to increase ciliary beating directly on the prototroch cells and indirectly by inhibiting the MC neuron (CBF 1 or CBF2) proportional to the change in pressure.

## Behavioural assays

### Design and assembly of pressure chambers

#### Large pressure chamber

The chamber's construction was based on an early design by *Lincoln and Gilchrist, 1970*. In brief, it consisted of a central core of 110 mm (*H*) × 90 mm (*W*) × 20 mm (*D*) with window openings on the lateral, front, and bottom sides. Each of the openings had a 12-mm crevice to accept custom-made acrylic plastic windows (Theodor Schmid GmbH, Tübingen). The windows were fitted into the crevices each with 3 mm-thick silicon rubber gaskets and fastened with aluminum plates and screws on top of them. The central working dimensions of the fully assembled chamber are 100 mm × 70 mm × 10 mm (70 ml). A 10-mm diameter threaded opening was bored at the back of the chamber to which a shutoff valve was connected. The design was produced using Blender (blender.org) and translated to a proprietary CAD-based software (eMachineShop CAD, Version 1.82 (2013), Micro Logic Corp, USA; Blender file, EMS file). The chamber was made of anodized aluminum alloy 6061 T6 using CNC machining techniques (eMachineShop, USA). The final threading was made in a workshop (Hans Mast GmbH, Tübingen).

#### Small pressure chamber

A small chamber was used for experiments with fewer larvae per batch. This chamber was also utilized for the ciliary dynamics experiments (see below). The design followed that of the big chamber, except that this chamber had acrylic windows on all sides. The central working dimensions of the fully assembled chamber are 40 mm × 30 mm × 10 mm (12 ml). It was machined in aluminum by the EMBL mechanical workshop in Heidelberg.

## Behavioural experiments

A mix of age-matched batches of larvae from different crosses were added to the pressure chamber. The chamber was not filled completely to minimize water disturbances upon the addition of air. Bubbles adhered to the edges and walls of the chamber were removed with a plastic mixer. Behaviour was recorded in effective darkness with near-infrared LED stripes (850 nm, Cat#: 15412085A9000 Wurth Elektronik, Germany) placed on each side of the chamber. A CCD camera (DMK31BF03 or DMK41BF02, Imaging source, Germany) was placed in front of the chamber at a distance sufficient to capture the entire behavioural arena in the field of view. The camera was triggered in most experiments at 10 fps via TTL communication with an Arduino microcontroller (Arduino Uno R3, Arduino, Italy).

The chamber was connected to a compressed air tank (Zero Grade Air, Hydrocarbons R&A, 270029-AZ-C, BOC) using locally sourced fittings and hoses. The flow of air into and out of the chamber was controlled with a pair of solenoid valves (SV, Cat# 293478, Burkert, Germany, and ST-DA018S030F-012DC, JP Fluid Control, The Netherlands). Pressure was recorded across each trial with either a 50, 100, 250, or 1000 mb pressure transducer (Cat# 8285701, 8285713, 8285726, 7975043, RS-PRO, UK) with a voltage output linearly related to pressure. The pressure reading feedbacks to the SVs via the Arduino to maintain a constant pressure level, updating the reading faster than the acquisition speed (Arduino scripts used: sketch_WorkingScript_1_twovalvesNOavgUVNOvalvedelayLogicPressur.ino for the 50, 100, and 250 mb transducers and PressureProgram_1000 mb.ino for the 1000 mb transducer). For the longer-term pressure adaptation experiments, the same setup was used, but a voltage equivalent to 100 or 500 mb was used as basal pressure and kept constant. A manual proportional valve downstream of the solenoid valve was used to adjust the rate of linear increase in pressure.

To increase pressure using a column of water, a flexible hose filled with sea water was screwed to the pressure chamber pre-loaded with the larvae under water to avoid introducing air into the system. The hose had ball valves at both ends (*Figure 1—figure supplement 2C*). The valve closer to the chamber was closed after connecting the hose to the chamber. The hose was straightened to the desired height (an equivalence of 1 cm to 1 mb was assumed), and the upper valve was opened at this moment. To increase pressure inside the chamber the lower valve was opened. To release pressure the hose was lowered to the level of the chamber. Pressure levels between 10 and 100 mb with 10 mb increments were chosen and applied in a randomized order.

## Stimulation protocol

The general protocol consisted of recording the vertical swimming behaviour of larvae for 10 s before pressure was increased as a step function of time. Pressure was held at a set value for a defined amount of time; stimulus duration was in most cases 60 s. A list of target pressure values or rates of pressure increase/decrease were applied to the same group of larvae in randomized order (List randomization, random.org). A resting period of 5–10 min was allowed between trials.

## Data analysis

Tracks of swimming larvae (in *XY* coordinates) were extracted from the recordings using a modified version of the script by *Gühmann et al., 2015* (BatchBehaviourTrackExtraction.ijm). Each recording was split into 1-s fragments (in most cases 10 frames long), from which tracks spanning the full second were obtained with the MTracK2 plugin (Nico Stuurman) and formatted into a table using R (ReadMtrack2files.R).

The track coordinates were used to calculate a range of metrics (script MeasuresTracks.R) according to the formulae in *Table 1*.

A maximum of 400 tracks per file were analysed. Only data containing more than 50 tracks for single-batch experiments, or by 100 or more tracks for experiments using a mix of multiple batches were used for analysis.

Where indicated, results were averaged across trials, and the value normalized relative to average displacement prior to stimulus onset. Maximal values were obtained from smoothed triangular averages (i.e. a single moving average calculated twice, each with a window size of 5). For the long-exposure experiments, the percent increase in pressure at a time point $t$ was calculated as follows: $\% \, pressure, increment_t = \frac{(pressure_t + 1000) * 100}{pressure_{prior} + 1000 - 100}$ assuming the ambient pressure was ~1000 mb.

Larval distribution along the vertical axis of the pressure vessel was obtained by dividing the image into 5 bins of equal size and counting the number of particles (after thresholding) in each bin. The script ScriptParticleCounter.ijm was used for that purpose.

## Ciliary dynamics experiments

### Behavioural setup

To measure the ciliary dynamics of the main ciliary band (prototroch), 2-day-old larvae were tethered from the posterior end to a glass cuvette (15 mm × 15 mm × 4 mm) with non-toxic glue (Wormglu, GluStitch Inc). The cuvette was inserted into a custom-made three-dimensional printer adaptor and fitted into an air-tight vessel (small chamber, see above). The vessel was connected to the Arduino-controlled system of solenoid valves and electronic transducer used for the batch experiments. For experiments with injected larvae, fluorescein and reporter signals were confirmed prior to the experiment, but after gluing the larva to the cuvette. Randomized step increases in pressure were applied with a resting interval of 10 min between trials. Larvae in the device were imaged with an Axio-Zoom V.16 (Carl Zeiss Microscopy, Germany) fitted with a ×2.3 objective under effective darkness (i.e. filtering the light from a CL9000 LED lamp with a long-pass filter, Cat# 435700-9025-000, Carl Zeiss Microscopy, Germany). Recordings were acquired at 200 Hz with an ORCA Flash 4.0 camera (C11440-22CU, Hamamatsu) using µManager (*Edelstein et al., 2010*).

### CBF measurements

Kymographs of the entire ciliary band obtained with an ImageJ macro (CBFrollingAvgSubKimoUnsup.ijm) were used to measure the CBF and the beating state across each trial. The CBF was calculated by adjusting a ridge wave to the wavelet transform of the signal using the ridge extraction mode (band-pass filter: 5–22 Hz) of the MODA numerical toolbox (*Iatsenko et al., 2019*; *Iatsenko et al., 2016*), implemented with a Matlab script (batchprocessKymographs.m). A Forward Fourier Transform (FFT) for each time unit was also calculated from the data with the ImageJ macro, but it was not adequate to capture the time-varying frequency in the data. Nonetheless, the FFTs provided an indirect way of identifying ciliary arrests. The following formula was used to measure CBF from the FFTs:

$$CBF = FPS/R$$

where *FPS* is the recording rate and *R* is the radius value in the FFT power spectrum image. Ciliary arrests corresponded to *R* values smaller than 8 or greater than 17 (corresponding to CBF between 11.8 and 25 for 200 fps). An R script (CBFcalculation.R) code was used to integrate pressure logs of each trial and the CBFs measured from wavelet and Fourier transforms, as well as to record the beating state of the band for each second of each trial.

The CBF extracted from the wavelet transforms were averaged using a triangular moving average. The maximum CBF (max. CBF) from these smoothed data obtained for each period (i.e. before, during and after stimulus) of each trial was used for measuring significant changes in CBF. The percentage change in max.CBF during the stimulus period, %ΔCBF, was defined as follows: $\% \, \Delta CBF = \left( \frac{CBF - meanCBF_{prior}}{meanCBF_{prior}} \right) \times 100$ , where $meanCBF_{prior}$ is the average CBF (after applying the rolling average) before stimulus onset.

## Calcium imaging

### Imaging setup

Two- or three-day-old larvae injected with a mix of *GCaMP6s* (~1 µg/µl) and *Palmitoylated-tdTomato* mRNA (<0.2 ng/µl; see Microinjection and generation of plasmid constructs and mRNA section) were embedded in ~2.7% low-melting agarose (Hampton Research, Cat# HR8-092) with an anterior to posterior orientation and placed on a ø25 mm coverslip (Epredia, Cat# CB00250RAC33MNZ0). A water-tight enclosure around the agarose pad was adhered with Vaseline Jelly to the coverslip. ~120 µl fASW were added to the enclosure to keep the larva under water during the experiment. The preparation was inserted in the central cavity of a custom-made acrylic chamber (STEP file and Mechanical drawing file) and firmly held with an aluminium disk (~3 mm thick) screwed into the chamber. A silicone gasket was used to reduce air leakages. The chamber's pipeline was plugged into the compressed air system shown in *Figure 1A* to control pressure levels.

### Image acquisition and analysis

Z-stacks and TL recordings were acquired with an inverted Zeiss LSM880 confocal microscope with the 488 nm argon laser set at 1% intensity, and the He 543 laser at 0.5% with a ×40 water objective (421767-9971-790, Carl Zeiss Microscopy). Z-stacks of the entire larva were acquired immediately before, during and after increasing pressure to ~750 mb. TLs were acquired at ~4.85 Hz for 120 s. Randomized step increases in pressure were applied for 60 s after recording basal activity for 30 s. The change in focus in TL recordings induced by the increase in pressure was corrected by automatically shifting the recording plane to the predicted new focal plane as soon as pressure started increasing, followed by stabilization of the imaging plane with an infrared light-based system (DefiniteFocus2, Carl Zeiss Microscopy). The same procedure was applied when pressure was brought back to ambient levels.

Z-stacks acquired before, during, and after stimulus were aligned with Fijiyama (*Fernandez and Moisy, 2021*) using the tdTomato signal as a reference. This strategy was also used to correlate cells in Z-stacks before and after immunostaining (but using isotropic deformation). *X–Y* shifts in TL recordings were corrected using descriptor-based series registration (*Preibisch et al., 2010*), followed by measurement of the signal intensity in user-defined Regions of Interest (ROIs) across the TL with an ImageJ script (ExtractionIntensityValsTomGC.ijm). A corrected change in fluorescence Δ*R/R* (*Bezares-Calderón et al., 2018*; *Böhm et al., 2016*) based on both the GCaMP6s and the tdTomato signals was used to quantify neuronal activity (implemented in GCaMP_TLanalysis.R).

## Microinjection and generation of plasmid constructs and mRNA

*Platynereis* embryos were microinjected with mRNA or plasmid constructs following a modified version of the original protocol (*Ackermann et al., 2005*). *mRNA* was synthesized with the mMESSAGE mMACHINE T7 Transcription Kit (Ambion, Cat# AM1344) and eluted in RNAse-free water. The plasmid used to synthesize *GCaMP6s* mRNA was previously reported (pLB112, pUC57-T7-RPP2-GCaMP6-RPP2 in *Randel et al., 2014*).The *Palmitoylated tdTomato* (Palmi-tdTom) construct (pUC57-T7-RPP2-Palmi-3xHA-tdTom-RPP2 or pLB260) used to syntesize *Palmi-tdTom* mRNA was generated by subcloning the *Palmi-tdTom* ORF (previously reported in *Bezares-Calderón et al., 2018*) into the same vector used for *GCaMP6s* mRNA production.

**Table 1.** Swimming metrics.

| Metric | Formula |
| --- | --- |
| Average vertical displacement | $\dfrac{\sum_{j=1}^{Tracks}\sum_{i=1}^{Frames}\left(Y_{ji+1}-Y_{ji}\right)}{Tracks \times \#Frames} \times Framerate \times mmpx^{-1}$ |
| Vertical movement | $\dfrac{\sum_{j=1}^{Tracks}\frac{\left(Y_{jfin.pos}-Y_{jinit.pos}\right)}{Frames_j}}{Tracks} \times Framerate \times mmpx^{-1}$ |
| Average speed | $\dfrac{\sum_{j=1}^{Tracks}\sum_{i=1}^{Frames}\sqrt{\left(X_{ji+1}-X_{ji}\right)^2+\left(Y_{ji+1}-Y_{ji}\right)^2}}{Tracks \times Frames} \times Framerate \times mmpx^{-1}$ |
| Average straightness vertical path | $\dfrac{\sum_{j=1}^{Tracks}\sum_{i=1}^{Frames}\frac{\left(Y_{ji+1}-Y_{ji}\right)}{\sqrt{\left(X_{ji+1}-X_{ji}\right)^2+\left(Y_{ji+1}-Y_{ji}\right)^2}}}{Tracks \times Frames}$ |
| Straightness index | $\sum_{j=1}^{Tracks}\dfrac{\sqrt{\left(X_{jfin.pos}-X_{jinit.pos}\right)^2+\left(Y_{jfin.pos}-Y_{jinit.pos}\right)^2}}{\sum_{i=1}^{Frames}\sqrt{\left(X_{ji+1}-X_{ji}\right)^2+\left(Y_{ji+1}-Y_{ji}\right)^2}/Tracks}$ |

The TPHp::Palmi-tdTomato-P2A-TeTxLC (pLB316) was generated by subcloning the *TeTxLC* ORF amplified from pGEMTEZ-TeTxLC into a plasmid encoding the fused Palmi-tdTomato and the P2A self-cleaving peptide sequence pLB243 or pUC57-T7-RPP2-tdTomato-P2A-GCaMP6 in *Bezares-Calderón et al., 2018*, followed by sub-cloning the entire ORF downstream of the *TPHp* sequence (*Verasztó et al., 2017*). This plasmid or the control plasmid *TPHp::Palmi-tdTomato* pLB253, previously described; *Verasztó et al., 2017* were co-injected with 5 µg/ml dextran fluorescein (D1821) at 250 ng/µl in water. pGEMTEZ-TeTxLC was a gift from Richard Axel & Joseph Gogos & C. Ron Yu (Addgene plasmid # 32640; RRID:Addgene_32640).

## Whole mount immunochemistry

Whole mount immunostaining was carried out as previously described (*Verasztó et al., 2017*). Larvae were fixed in 4% formaldehyde made from a 16% stock (Electron Microscopy Sciences, Cat# 15710) diluted in Phosphate-Buffered Saline (PBS) (NaCl: 137 mM, KCl: 2.7 mM, $Na_2HPO_4$: 10 mM, $KH_2PO_4$: 1.8 mM) + 0.1% Tween for 15 min at room temperature (RT). Larvae used after $Ca^{2+}$ imaging and for assessment of the cPRC cilia volume were incubated overnight at 4°C with a polyclonal rabbit NIT-GC2 antibody at 5 µg/ml, and in some cases combined with a rabbit anti-5HT (serotonin) at 2 µg/ml (ImmunoStar Cat# 20080, RRID:AB_572263). Depending on the aim of the staining, larvae were counterstained with either a mouse monoclonal anti-HA tag antibody (Cell Signaling Technology, HA-Tag (6E2), Cat #2367, RRID:AB_10691311) or with a mouse monoclonal anti-acetylated tubulin antibody (Sigma-Aldrich, Cat# T6793, RRID:AB_477585) at 1:250 dilution. Plasmid-injected larvae were also counterstained with the anti-acetylated tubulin antibody and with a rabbit monoclonal anti-HA tag antibody (Cell Signaling Technology, HA-Tag (C29F4) Cat# 3724, RRID:AB_1549585). Samples were incubated overnight at 4°C with 1:250 dilutions of the anti-rabbit Alexa Fluor 488 (Thermo Fisher Scientific, Cat# A-11008, RRID:AB_143165), anti-mouse F(ab) fragment Alexa Fluor 546 (Thermo Fisher Scientific, Cat# A-11018, RRID:AB_2534085) secondary antibodies, and stained with 500 ng/ml 4',6-diamidino-2-phenylindole (DAPI) for 15 min at RT.

## cPRCs morphology assessment
### Light microscopy analysis
Z-stacks of larvae stained with the cPRC cilia antibody were acquired with equal acquisition settings and were processed with the ImageJ 3D Object Counter (Cordelires et al.) plugin through the VolumeCalc.ijm script. A threshold intensity that included the entire ciliary structure was manually adjusted for each scan, filtering out structures with less than 1000 voxels. The volumes thus obtained were imported and quantified in R (VolumecPRCanalysis.R).

## Electron microscopy analysis

### Sample preparation

*c-ops-1*$^{\Delta 8/\Delta 8}$ larvae used for serial scanning electron microscopy (SEM) were mounted in 20% Ficoll PM70 (Sigma, Cat# F2878) and high-pressure frozen (HPM Live-μ, Labtech). Samples were then dehydrated by freeze substitution using a standard procedure (*McDonald and Webb, 2011*) with modifications of the fixation solution (1% osmium tetroxide + 0.5% uranyl acetate (UA) + 0.5% glutaraldehyde (GA) + 3% $H_2O$). Fixed larvae were embedded in epoxy (EMBED 812/DMP-30, Cat# 14120, Electron Microscopy Sciences) and sectioned at ~70 nm using a microtome (UC7, Leica) equipped with a diamond knife (histo Jumbo 45° Cat# HI8592, DiATOME). Image acquisition was done by SEM (GeminiSEM 500, Zeiss) using an energy selective backscatter detector. Sections were elastically aligned with TrakEM2 (*Cardona et al., 2012*). cPRC cilia were reconstructed in CATMAID (*Saalfeld et al., 2009*).

This manuscript was written in RMarkdown (*Xie, 2015*).

## Acknowledgements

We thank Paulina Cherek for fixing and embedding samples for sSEM, Kei Jokura for kindly sharing the anti-cPRC cilia antibody, Cameron Hird, Adam Johnstone, and Rebecca Turner for animal husbandry, Sanja Jasek for IT support and data server maintenance, Rebecca Poon for advice on quantification of ciliary dynamics, and Emelie Brodrick and Elizabeth Williams for critical reading of the manuscript. This research was funded by the Wellcome Trust Investigator Award 214337/Z/18/Z and the Biotechnology and a Biological Sciences Research Council Response Mode BB/W00853X/1 and ALERT BB/S019499/1 Grant. This project has received funding from the European Research Council (ERC) under the European Union's Horizon 2020 research and innovation programme (grant agreement No 101020792).

## Additional information

### Competing interests

Gáspár Jékely: Reviewing editor, *eLife*. The other authors declare that no competing interests exist.

### Funding

| Funder | Grant reference number | Author |
| --- | --- | --- |
| Wellcome Trust | 214337/A/18/Z | Gáspár Jékely |
| Biotechnology and Biological Sciences Research Council | BB/W00853X/1 | Luis Alberto Bezares Calderón<br>Gáspár Jékely |
| European Research Council | 101020792 | Gáspár Jékely |

The funders had no role in study design, data collection, and interpretation, or the decision to submit the work for publication. For the purpose of Open Access, the authors have applied a CC BY public copyright license to any Author Accepted Manuscript version arising from this submission.

### Author contributions

Luis Alberto Bezares Calderón, Conceptualization, Resources, Data curation, Software, Formal analysis, Validation, Investigation, Visualization, Methodology, Writing - original draft, Writing - review and editing; Réza Shahidi, Resources, Investigation, Methodology; Gáspár Jékely, Conceptualization, Data curation, Software, Supervision, Funding acquisition, Visualization, Writing - original draft, Project administration, Writing - review and editing

### Author ORCIDs

Luis Alberto Bezares Calderón http://orcid.org/0000-0001-6678-6876
Gáspár Jékely https://orcid.org/0000-0001-8496-9836

Joint Public Review: https://doi.org/10.7554/eLife.94306.3.sa1
Author response https://doi.org/10.7554/eLife.94306.3.sa2

## Additional files

### Supplementary files
• MDAR checklist

### Data availability

The volume EM stack of the c-opsin1 mutant ciliary photoreceptors is available at https://catmaid.jekelylab.ex.ac.uk/ (Platynereis cPRCs c-opsin-1 mutant). Source data files have been provided in the submission containing the numerical data used to generate the figures.Code documentation for the paper has been deposited to Zenodo.

The following datasets were generated:

| Author(s) | Year | Dataset title | Dataset URL | Database and Identifier |
|---|---|---|---|---|
| Bezares Calderón LA, Shahidi R, Jékely G | 2023 | Platynereis cPRCs c-opsin-1 mutant | https://catmaid.jekelylab.ex.ac.uk | CATMAID, Platynereis cPRCs c-opsin-1 mutant |
| Bezares Calderón LA, Shahidi R, Jékely G | 2024 | Code documentation for Mechanism of barotaxis in marine zooplankton | https://doi.org/10.5281/zenodo.11103057 | Zenodo, 10.5281/zenodo.11103057 |

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
