## [Editor Report · eLife assessment]

This **fundamental** study addresses the question of how certain zooplankton achieve barotaxis, directed locomotion in response to changes in hydraulic pressure. The authors provide **compelling** evidence that the response involves ciliary photoreceptors interacting with motoneurons. This work should be of broad interest to scientists working on mechanosensation, cilia, locomotion, and photoreceptors.

---

## [Referee Report · Joint Public Review]

In this work, the authors address a fundamental question in the biological physics of many marine organisms, across a range of sizes: what is the mechanism by which they measure and respond to pressure. Such responses are classed under the term "barotaxis", with a specific response termed "barokinesis", in which swimming speed increases with depth (hence with pressure). While macroscopic structures such as gas-filled bladders are known to be relevant in fish, the mechanism for smaller organisms has remained unclear. In this work, the authors use ciliated larvae of the marine annelid *Platynereis dumerilii* to investigate this question. This organism has previously been of great importance in unravelling the mechanism of multicellular phototaxis associated with a ciliated band of tissue directed by light falling on photoreceptors.

In the present work, the authors use a bespoke system to apply controlled pressure changes to organisms in water and to monitor their transient response in terms of swimming speed and characteristics of swimming trajectories. They establish that those changes are based on relative pressure, and are reflected in changes in the ciliary beating. Significantly, by imaging neuronal activity during pressure stimulation, it was shown that ciliary photoreceptor cells are activated during the pressure response. That these photoreceptors are implicated in the response was verified by the reduced response of certain mutants, which appear to have defective cilia. Finally, serotinin was implicated in the synaptic response of those neurons.

This work is an impressive and synergistic combination of a number of different biological and physical probes into this complex problem. The ultimate result, that ciliary photoreceptors are implicated, is fascinating and suggests and interesting interplay between photoreception and pressure detection.

Future studies ought to address the following three questions opened by this work:

(1) How the off response to decrease of pressure is mediated

(2) Which receptor/channel mediates in photoreceptors the response to increased pressure,

(3) How the integration of light and pressure information is integrated by photoreceptors in order to guide the behavior of the larvae.

---

## [Author Response]

The following is the authors’ response to the original reviews.

**Reviewer #1:**

We thank Reviewer #1 for the assessment of our study.

**Reviewer #2:**
The authors should use DF/F to quantify over time the calcium response in photoreceptors. Furthermore, they should show that there is no concern of motion artifact when the pressure changes - as it could be a concern”.

We used the ΔR/R measure (as defined in Böhm et al. 2016) to correct for motion artifacts due to the larvae moving out of the focal plane at the onset of pressure stimulation. This measure calculates the ratio of the GCaMP signal and a reference fluorescent signal (tdTomato in our case). This ratiometric quantification can better correct for changes in fluorescence that are not related to changes in calcium concentration than the ΔF/F metric, which does not use an independent reference channel.

The authors have not shown(1) how the off response to decrease of pressure is mediated(2) which receptor/channel mediates in photoreceptors the response to increased pressure,(3) nor how the integration of light and pressure information is integrated by photoreceptors in order to guide the behavior of the larvae.These points are beyond the scope of the study. However, if possible within a short time frame, it would be really interesting to find out whether conflicting stimuli or converging stimuli (light & pressure) can cancel each other out or synergize. In particular since the authors cite unpublished results in the discussion: "Our unpublished results indeed suggest that green light determines the direction of swimming and can override upward swimming induced by pressure, which only influences the speed of swimming (LABC and GJ, unpublished)." Showing in one panel this very cool phenomenon would be exciting & open tons of questions for the field.”

We agree that investigating the interaction of light and pressure is a very exciting direction. However, doing it properly with the rigour we characterised pressure sensation here (across stages, pressure levels and genotypes) and phototaxis and UV avoidance in previous work (across stages, wavelengths, genotypes and stimulus direction; see Randel et al. 2014, Gühmann et al. 2015, Verasztó et al. 2018, Jokura et al. 2023) would require a separate in-depth study.

We agree with points 1-3 regarding the limitations and mentioned these in the discussion.

(1) Although we carried out pressure-release experiments to characterise in more detail the response to pressure OFF, our setup did not allow us to control pressure release as accurately as we could for pressure increase. Therefore, we decided not to address this aspect of the response in more detail in this study.

“Upon a decrease in pressure, three-day-old (but not two-day-old) larvae also show an off-response characterised by downward swimming. We have not analysed in detail the neuronal mechanisms of this response but it may depend on an inverted activation of the cPRC circuit, as happens during UV avoidance (Jokura et al., 2023)”

(2) We decided not to explore this important question in this study, due to the significant effort it would take to test the expression and function of potential candidate channels in pressure transduction mechanism. “The cellular and molecular mechanisms by which cPRCs sense and transduce changes in hydrostatic pressure deserve further enquiry. “ and “The molecular mechanisms of pressure detection remain unclear. Components of the phototransduction cascade may be involved in pressure sensation. Our results indicate that the ciliary opsin required for detecting UV light is not essential for pressure sensation.“ We hypothesise in the discussion that TRP channels may play a role in pressure transduction, due to their diversity, multiple modalities and participation in phototransduction cascades.

(3) We considered that the complexity of this question merits a separate study, where both cues can be accurately titrated and temporally combined to dissect the mechanisms of sensory integration. We have therefore removed the sentence referring to the interaction of phototaxis and the pressure response from the discussion.

“How UV and pressure signals are integrated by the cPRC and how other light responses such as phototaxis interact with pressure responses remain exciting avenues for future research.”